# MicroRNAs Associated with the Pathophysiological Mechanisms of Gestational Diabetes Mellitus: A Systematic Review for Building a Panel of miRNAs

**DOI:** 10.3390/jpm13071126

**Published:** 2023-07-11

**Authors:** Pedro Henrique Costa Matos da Silva, Kamilla de Faria Santos, Laura da Silva, Caroline Christine Pincela da Costa, Rodrigo da Silva Santos, Angela Adamski da Silva Reis

**Affiliations:** 1Laboratory of Molecular Pathology, Institute of Biological Sciences, Federal University of Goiás (UFG), Goiânia 74690-090, GO, Brazilkamillafaria@discente.ufg.br (K.d.F.S.);; 2Department of Biochemistry and Molecular Biology, Institute of Biological Sciences, Federal University of Goiás (UFG), Goiânia 74690-090, GO, Brazil

**Keywords:** GDM, diabetes, miRNA, personalized medicine

## Abstract

miRNAs, a class of small non-coding RNAs, play a role in post-transcriptional gene expression. Therefore, this study aimed to conduct a systematic review of miRNAs associated with GDM to build a panel of miRNAs. A bibliographic search was carried out in the PubMed/Medline, Virtual Health Library (VHL), Web of Science, and EMBASE databases, selecting observational studies in English without time restriction. The protocol was registered on the PROSPERO platform (number CRD42021291791). Fifty-five studies were included in this systematic review, and 82 altered miRNAs in GDM were identified. In addition, four miRNAs were most frequently dysregulated in GDM (mir-16-5p, mir-20a-5p, mir-222-3p, and mir-330-3p). The dysregulation of these miRNAs is associated with the mechanisms of cell cycle homeostasis, growth, and proliferation of pancreatic β cells, glucose uptake and metabolism, insulin secretion, and resistance. On the other hand, identifying miRNAs associated with GDM and elucidating its main mechanisms can assist in the characterization and definition of potential biomarkers for the diagnosis and treatment of GDM.

## 1. Introduction

Gestational Diabetes Mellitus (GDM) is defined as any degree of glucose intolerance first diagnosed during pregnancy [1]. Hyperglycemia during pregnancy can be transient or persist after birth, presenting itself as an independent risk factor for the future development of Type 2 Diabetes Mellitus (T2DM) [2]. According to the International Diabetes Federation, the global prevalence of GDM averages 14% [3], ranging from 1.8–31% depending on the population evaluated and the diagnostic criteria adopted between countries [4]. GDM is the most common metabolic disease during pregnancy, occurring in 3–25% of pregnancies, and its incidence in the population has increased along with T2DM and obesity [5].

There is no universally accepted standard for screening or diagnosing GDM. Guidelines from local medical organizations are followed. However, the test with better sensitivity and specificity is the oral glucose tolerance test (OGTT) with 75 g of glucose, considering values between 153 and 199 mg/dL for GDM [6]. GDM diagnosis can be performed throughout pregnancy from the beginning of prenatal care. However, it is usually performed in the second or third trimester of pregnancy (24–28 weeks), which can cause risks to the mother and the fetus [7]. Short-term adverse outcomes are observed, such as hypoglycemia, hypoxia, respiratory distress syndrome, higher rates of preeclampsia, and large for gestational age or macrosomic newborns, among others [2,5,8]. In addition, children born from mothers with GDM have an increased risk for metabolic and cerebrovascular diseases in adult age [5].

Pregnancy stresses the body and promotes physiological changes to ensure the proper growth of the embryo/fetus. Adaptations and/or dysregulations during pregnancy are performed by placental hormones and increased levels of cortisol and progesterone [9]. Additionally, it is reported that molecules such as placenta-derived microRNAs (miRNAs) may be involved in these adaptations. Variations in the expression of these miRNAs may indicate changes in the maternal metabolic adaptation mechanism [7,10].

miRNAs are a class of endogenous non-coding RNAs with approximately 22 nucleotides, which act as regulators of post-transcriptional gene expression, inhibiting the translation of messenger RNAs (mRNA) or degrading them [5,11]. miRNAs regulate more than one target mRNA, and studies previously described that occurs a control of the expression to an average of 30% of protein-coding genes [7,12]. The main well-known functions of miRNAs are the regulation of cell proliferation and differentiation, apoptosis, stress response, and transcriptional regulation [7].

Studies indicate that the human placenta expresses more than 500 miRNAs, and only some of these are also expressed in other tissues [13]. Therefore, the characterization of miRNAs during pregnancy is necessary to understand better the regulatory mechanisms of healthy and complicated pregnancy [14]. Currently, studies have sought to identify biomarkers for the diagnosis of GDM before 24–28 weeks of gestation, and these miRNAs have revealed great potential as biomarkers for GDM in the early trimester, mainly due to their high stability and accessibility in body fluids [15,16]. 

Additionally, investigating the regulation of placental and circulating miRNAs and their metabolic adaptation associated with GDM can improve diagnostic, therapeutic, and personalized prognosis [7]. Thus, understanding the functions of miRNAs can improve broadening the insight into the etiology and pathophysiology of GDM and identify possible biomarkers with clinical value to elaborate diagnostic strategies and prevent obstetric and maternal–fetal complications. 

## 2. Materials and Methods

### 2.1. Registration, Data Source, and Search Strategy

Based on the guiding question, “Which microRNAs are associated with the pathophysiological mechanisms of Gestational Diabetes Mellitus (GDM)?”, the systematic review was carried out to identify the miRNAs associated with the pathophysiological mechanisms of GDM. We used the PEO (Population, Exposure, Outcome) framework: Population: pregnant woman with GDM; Exposure: microRNAs; Outcome: association of microRNAs as a risk factor for GDM. 

This systematic review was carried out according to the Preferred Report Items for Systematic Reviews and Meta-Analyses (PRISMA) guidelines [17]. To avoid duplication, this study had the protocol registered in the International Prospective Register of Systematic Reviews (PROSPERO) (number CRD42021291791) on 17 December 2021 (Appendix A).

The literature search was carried out in PubMed/Medline, Virtual Health Library (VHL), Web of Science, and EMBASE databases from 27 January 2022 to 15 February 2022 in the English language. Combined terms cataloged in Medical Subject Headings (MeSH) for the keywords (Gestational Diabetes OR Gestational Diabetes Mellitus OR Diabetes Induced by Pregnancy) AND (MicroRNA OR miRNA OR Circulating MicroRNA OR Cell-Free MicroRNA) were used to develop the search strategy, later adapted for each database (Table 1).

As criteria for inclusion of studies in this systematic review, we used: observational studies with original research in the area of human and medical genetics that identified miRNAs in pregnant women with GDM and controls (pregnant women without GDM); no age restrictions; published in English; without the restriction of the year of publication of the study. For exclusion, the following criteria were established: Studies that did not address the research topic; and with another study design.

### 2.2. Selection of Studies

Two reviewers (PHCMS and LS) independently selected and identified articles at all stages of the systematic review. The articles identified in the databases were imported into the Rayyan platform [18] to optimize the analysis. Initially, the title and abstract were read (phase I), then the articles selected in the first screening were read in full (phase II). In both phases, the articles were evaluated according to the pre-established inclusion and exclusion criteria. 

### 2.3. Risk of Bias Assessment

The risk of methodological bias was assessed using The Joanna Briggs Institute (JBI) Critical Assessment Tool [19] for each study design. All articles selected for inclusion in the systematic review underwent rigorous evaluation. This tool is composed of questions with the possible answers: “Yes”, “No”, “Not applicable”, and “Unclear”.

A critical assessment tool was applied to each type of study: case-control studies, cohort studies, and cross-sectional studies. Those articles that had 100% of the answers “yes” were considered at low risk of bias; those who scored 70–99% “yes” were considered moderate risk; and those with less than 70% of “yes” answers were excluded from the study, being considered at high risk of methodological bias. Differences of opinion between reviewers were discussed and resolved by consensus.

### 2.4. Data Extraction and Synthesis

Studies evaluating miRNAs lack more robust statistical analyses; most describe only the type of regulation identified and whether there was a significant difference in this regulation between the evaluated groups (usually considering *p* < 0.05). Thus, this systematic review used a qualitative and descriptive approach for data analysis, extracting the following data: (1) authors and year of publication; (2) study design; (3) country or continent where the study was performed; (4) sample size; (5) mean age; (6) gestational time; (7) tissue analyzed; (8) microRNA identification technique; (9) microRNA; (10) regulation; (11) *p*-value. The extracted data were imported into a predefined Excel worksheet, and authors of studies with missing data were contacted. Data extraction was performed independently by two reviewers, and disagreements between reviewers were resolved by consensus.

## 3. Results

After the initial search, 1138 articles were identified in the databases, and 626 duplicates were excluded, resulting in 512 articles for phase I. Based on the reading of titles and abstracts (phase I), 413 articles were excluded because they did not meet the pre-established inclusion criteria. Therefore, phase II comprised the complete reading of the 99 selected articles. Forty-four were excluded, and 55 were included in this systematic review (Figure 1).

Of the included studies, 48 were case-control studies, six were cohort studies, and one was a cross-sectional study. The selected studies were published between 2011 and 2022. The mean age of the case and control groups ranged between 20 and 40 years. A total of 2749 cases (pregnant women with GDM) and 2710 controls (pregnant women without GDM) were analyzed, totaling 5459 individuals. The samples collected and analyzed were blood, plasma, placenta, adipose tissue, and urine (urinary exosome) (Table 2).

Additionally, the studies were homogeneous in terms of methodological quality assessment, individually reaching a minimum of 70% of positive responses, being considered at low or moderate risk of bias (Table 3). For case-control studies, 10 parameters were evaluated, cohorts 11, and cross-sectional studies 8 parameters, respectively. Most case-control studies reported that there was no identification of possible confounding factors (question 6—Q6), and, consequently, the non-applicability of question 7 (Q7) declared strategies to deal with confounding factors.

In this systematic review, 82 miRNAs were identified, highlighting the most cited in the literature due it was deregulated in GDM (mir-16-5p, mir-20a-5p, mir-222-3p, and mir-330-3p). According to the results, the identified miRNAs were down- and/or up-regulated in the GDM, considering *p* < 0.05 as significant. Studies evaluating miRNAs lack more robust statistical analyses, often describing whether there was a significant difference and the type of miRNA regulation between the evaluated groups.

Among those most frequently found in the literature, mir-16-5p [40,47,68,70] and mir-330-3p [28,39,50] were up-regulated, and mir-20a-5p and mir-222-3p were up- and down-regulated in different studies [40,51,62,68,70,72] (Figure 2). In general, the dysregulation of these miRNAs is associated with mechanisms of inflammation, growth, and proliferation of pancreatic β cells, glucose uptake and metabolism, insulin secretion, and resistance (Table 4).

## 4. Discussion

miRNAs have been studied as potential biomarkers of GDM, and knowledge about their regulation and function, as well as metabolic adaptation associated with GDM, have been investigated and may help in the understanding of the pathogenesis of the disease [7]. In this systematic review, we identified 82 altered miRNAs in GDM. Of these, four were most cited, mir-16-5p and mir-330-3p were up-regulated, and mir-20a-5p and mir-222-3p were found to be up-regulated and down-regulated, respectively.

According to Gao and Zhao [75], mir-16-5p controls cell proliferation, migration, and invasion, affecting the cell cycle and promoting apoptosis. In addition, this miRNA modulates the PI3K/Akt signaling pathway, an important cell cycle regulator that includes genes such as Pi3Kr1, Pi3kr3, mTOR, and Mapk3, among others. Kwon et al. [76] observed that the up-regulation of mir-16-5p in the liver of Cmah-null mice, used as models for T2DM, can negatively regulate the insulin/PI3K-AKT signaling pathway in association with other genes. These results suggest that the impairment of insulin mechanisms may favor the development of metabolic disorders, such as chronic hyperglycemia. 

Other targets of this miRNA are the genes Insulin Receptor Substrate 1 and 2 (IRS1/IRS2) and Insulin-Like Growth Factor 1 (IGF-1), which are closely related to insulin resistance, a characteristic condition of diabetes [68,76]. In addition, the up-regulation of this miRNA in GDM patients in the second trimester of gestation has demonstrated a correlation with the down-regulation of IRS1 and IRS2, leading to abnormal signaling of the Wnt/β-catenin pathway, important for embryonic development and adult tissue homeostasis [77].

Whereas mir-330-3p is associated with proliferation, differentiation, and insulin secretion and is highly expressed in GDM related to high glucose concentration. It also acts as a central regulator of cell cycle homeostasis. Thus, the up-regulation of this miRNA in patients with GDM may contribute to pancreatic β cell dysfunction, altering the proliferation and growth of these cells [39,50]. On the other hand, studies have revealed that the E2F Transcription Factor 1 (E2F1) and Cell Division Cycle 42 (CDC42) genes are targets of mir-330-3p. Both are involved in the growth and proliferation of β cells and the control of insulin secretion. Therefore, the low expression of these genes is caused by increased miRNA levels that can compromise β cell proliferation and insulin secretion [39,50].

Moreover, the Angiotensin II receptor Type 2 (AGTR2) gene was also identified as a target of this miRNA; this gene acts during the development of the pancreas in the embryonic stage and is a possible mediator of the regeneration of β cells in the adult pancreas. Thus, it is hypothesized that elevated levels of mir-330-3p may inhibit pancreatic neogenesis through the down-regulation of AGTR2, causing a defect in the regeneration of the endocrine pancreas under high metabolic demand [50].

Additionally, in this systematic review, Zhu et al. [40] and Cao et al. [68] identified up-regulated mir-20a-5p, while Pheiffer et al. [62] found it down-regulated in GDM. Mir-20a-5p belongs to the mir-17-92 cluster and is associated with angiogenesis [62]. In GDM, the expression of angiogenic proteins is increased, such as Vascular Endothelial Growth Factor-A (VEGFA), Hypoxia-inducible factor 1 subunit alpha (HIF1A) [78], Phosphatase and Tensin homolog (PTEN) [79], and BCL2 apoptosis regulator (BCL2) [80]. This fact corroborates the regulatory effect of the decreased expression of mir-20a-5p found by Pheiffer et al. [62].

Zhu et al. [40] also associated mir-20a-5p with the insulin, MAPK, TGF-β, and mTOR signaling pathways. The PI3K/Akt pathway that regulates the cell cycle, glucose homeostasis, and insulin signaling, as well as the FoxO protein, which also regulates the insulin/PI3K/Akt pathway, were considered targets of mir-20a-5p. Thus, this miRNA becomes a possible biomarker of GDM, and the alteration of its expression can interfere with these pathways, generating hyperglycemia observed in GDM patients [62].

MAPK signaling pathway plays a role in developing vascular lesions, such as diabetes [81], and its abnormal signaling has been identified in pregnancy complications [40]. In addition, the TGF-β signaling pathway has been associated with preeclampsia [82]. In contrast, the mTOR signaling pathway controls energy balance [83] and blocking these pathways may contribute to the development of GDM [40].

Finally, mir-222-3p was identified by Tagoma et al. [72] and Filardi et al. [51] found it up-regulated, while Oostdam et al. [70] and Pheiffer et al. [62] found it down-regulated in GDM. Mir-222-3p belongs to the mir-222 cluster, is abundant in plasma during 24–28 weeks of gestation, and is a placental miRNA that acts on the proliferation of endometrial stromal cells [51,84], regulating the expression of estrogen receptor-α (ER-α) in estrogen-induced insulin resistance in GDM [55,62], and is strongly linked to glucose metabolism in pregnancy, profoundly impacting in the weight birth [51]. 

Furthermore, high levels of this miRNA were found in the adipose tissue of GDM patients, negatively correlated with the levels of ER-α and glucose transporter type 4 (GLUT4) [55]. Women with GDM have higher levels of estradiol when compared to healthy women, and the estradiol and ER-α act on the GLUT4, becoming critical regulators of obesity and insulin resistance [51].

Due to the relation between miRNAs and the regulation of gene expression, information about the tissue origin of malignant cells could be generated. Since their initial discovery, it has been clear that miRNAs are expressed in a variety of cell types and that their expression patterns are tissue-specific and thus could have great diagnostic importance and prognostic value [85].

The placenta produces several miRNAs expressed specifically by placental cells. They can be dysregulated in the plasm and placenta of women with GDM, being also associated with pregnancy and birth-related outcomes. Detection of placental miRNAs in placental cells and circulation contributes to the understanding of the molecular pathways and intracellular signalization and their influences on GDM genetic background [86]. Therefore, identifying possible GDM biomarkers has a significant clinical value in developing diagnostic strategies and possibly preventing obstetric and maternal–fetal complications [7]. However, studies on early recognition, diagnostic criteria, possible biomarkers, and therapeutic targets for GDM show controversial results, mainly due to differences in ethnic, geographic, genetic, and environmental factors and diagnostic criteria [7,87,88].

## 5. Conclusions

A miRNA can regulate up to 200 mRNAs, indicating that miRNAs can individually regulate many different biological processes [52]. Thus, it is suggested that miRNAs be used as biomarkers in miRNA panels or risk assessment algorithms and not as individual biomarkers for diagnosing GDM [62]. In this systematic review, we found 55 articles that analyzed the alteration of miRNAs in GDM, identifying 82 altered miRNAs. The expression of these miRNAs varied depending on the type of sample used and the different gestational ages, which may explain the differences in the results. Additionally, these miRNAs were associated with several mechanisms, such as cell cycle homeostasis, growth, and proliferation of pancreatic β cells, glucose uptake and metabolism, insulin secretion, and resistance. Certainly, miRNAs are potential biomarkers for the early diagnosis of GDM. However, more research is needed to elucidate their diagnostic and predictive value and their pathogenic mechanisms in pregnancy.

## Figures and Tables

**Figure 1 jpm-13-01126-f001:**
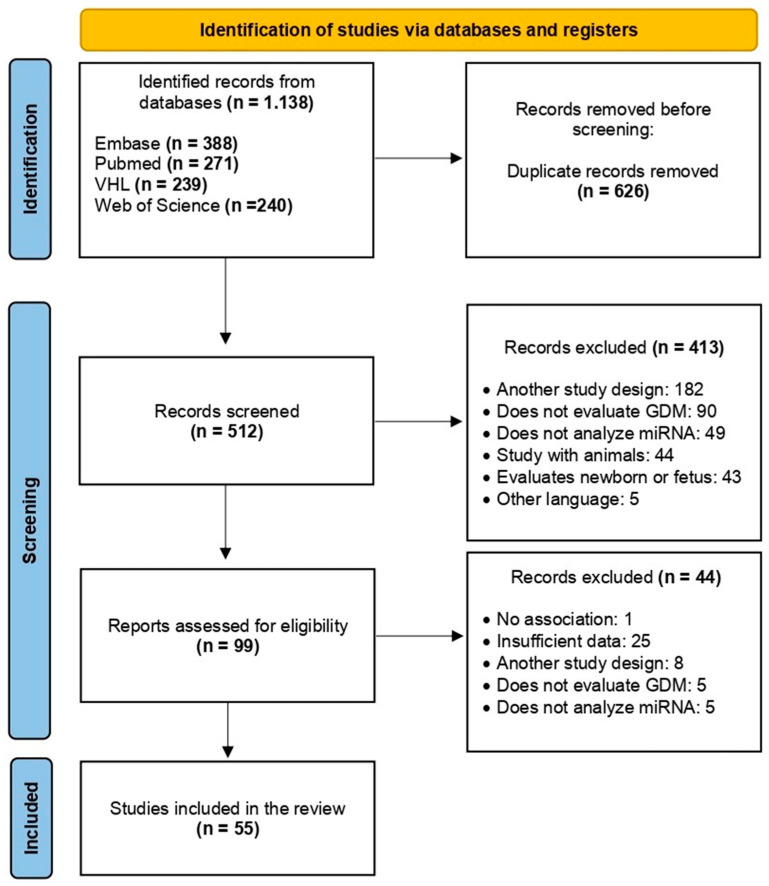
PRISMA flow chart demonstrating the process of exclusion or inclusion of studies in this systematic review. Adapted from: [17].

**Figure 2 jpm-13-01126-f002:**
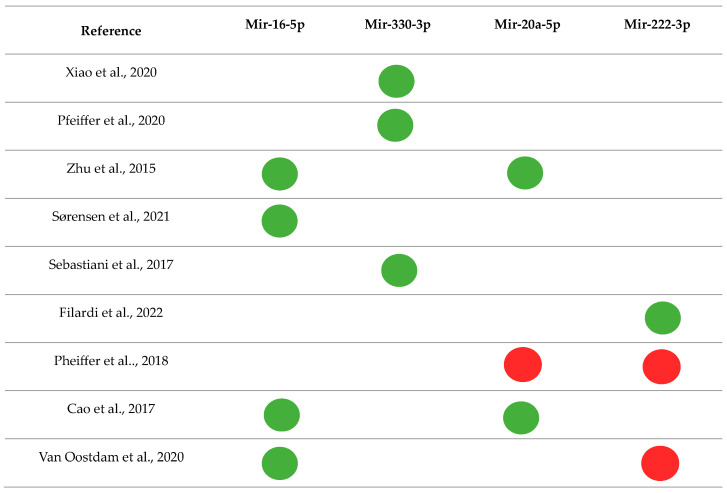
Main miRNAs and their regulations were found in the systematic review [28,39,40,47,50,51,62,68,70,72].

**Table 1 jpm-13-01126-t001:** Search strategy applied in each database.

Database	Search Strategy
PubMed/Medline	((“Diabetes, Gestational” OR “Diabetes Gestacional” OR “Diabetes Mellitus, Gestational” OR “Diabetes, Pregnancy Induced” OR “Diabetes, Pregnancy-Induced” OR “Gestational Diabetes” OR “Gestational Diabetes Mellitus” OR “Pregnancy-Induced Diabetes”) AND (“MicroRNAs” OR “Micro RNA” OR “MicroRNA” OR “MicroRNA, Primary” OR “miRNA” OR “miRNA, Primary” OR “miRNAs” OR “pre miRNA” OR “pre-miRNA” OR “pri miRNA” OR “pri-miRNA” OR “Primary MicroRNA” OR “Primary miRNA” OR “RNA, Micro” OR “Small Temporal RNA” OR “stRNA” OR “MicroARNs” OR “miARN” OR “miARNs” OR “Micro ARN” OR “Micro ARNs” OR “MicroARN” OR “stARN” OR “Circulating MicroRNA” OR “Cell Free MicroRNA” OR “Cell-Free MicroRNA” OR “MicroRNA, Cell-Free” OR “MicroRNA, Circulating”))
Virtual Health Library	(“microrna”) AND (“gestational diabetes”) AND (mj:(“Diabetes Gestacional” OR “MicroRNAs”))
Web of Science	((“Diabetes, Gestational” OR “Diabetes Mellitus, Gestational” OR “Diabetes, Pregnancy Induced” OR “Diabetes, Pregnancy-Induced” OR “Gestational Diabetes” OR “Gestational Diabetes Mellitus” OR “Pregnancy-Induced Diabetes”) AND (“MicroRNAs” OR “Micro RNA” OR “MicroRNA” OR “MicroRNA, Primary” OR “miRNA” OR “miRNA, Primary” OR “miRNAs” OR “pre miRNA” OR “pre-miRNA” OR “pri miRNA” OR “pri-miRNA” OR “Primary MicroRNA” OR “Primary miRNA” OR “RNA, Micro” OR “RNA, Small Temporal” OR “Small Temporal RNA” OR “sirna” OR “stRNA” OR “Temporal RNA, Small” OR “MicroARNs”))
EMBASE	(‘diabetes, gestational’/exp OR ‘diabetes, gestational’ OR ‘diabetes gestacional’ OR ‘diabetes, pregnancy induced’ OR ‘diabetes, pregnancy-induced’ OR ‘gestational diabetes’/exp OR ‘gestational diabetes’ OR ‘gestational diabetes mellitus’/exp OR ‘gestational diabetes mellitus’ OR ‘pregnancy-induced diabetes’) AND (‘micrornas’/exp OR ‘micrornas’ OR ‘micro rna’/exp OR ‘micro rna’ OR ‘microrna’/exp OR ‘microrna’ OR ‘microrna, primary’ OR ‘mirna’/exp OR ‘mirna’ OR ‘mirna, primary’ OR ‘mirnas’/exp OR ‘mirnas’ OR ‘pre mirna’ OR ‘pre-mirna’ OR ‘pri mirna’ OR ‘pri-mirna’ OR ‘primary microrna’/exp OR ‘primary microrna’ OR ‘primary mirna’ OR ‘rna, micro’ OR ‘rna, small temporal’ OR ‘small temporal rna’ OR ‘strna’ OR ‘temporal rna, small’ OR ‘microarns’ OR ‘arn pequeño temporal’ OR ‘arn temporal pequeño’ OR ‘miarn’ OR ‘miarns’ OR ‘micro arn’ OR ‘micro arns’ OR ‘microarn’ OR ‘microarn primario’ OR ‘microrna primario’ OR ‘starn’ OR ‘rna pequeno temporário’ OR ‘rna temporário pequeno’ OR ‘circulating microrna’/exp OR ‘circulating microrna’ OR ‘cell free microrna’/exp OR ‘cell free microrna’ OR ‘cell-free microrna’/exp OR ‘cell-free microrna’ OR ‘microrna, cell-free’ OR ‘microrna, circulating’ OR ‘microarn circulante’ OR ‘microrna circulante’ OR ‘microrna fora da célula’ OR ‘microrna livre’) AND (‘biological marker’/dd OR ‘circular ribonucleic acid’/dd OR ‘glucose’/dd OR ‘insulin’/dd OR ‘microrna’/dd OR ‘microrna 122’/dd OR ‘microrna 143’/dd OR ‘microrna 146a’/dd OR ‘microrna 16’/dd OR ‘microrna 21’/dd OR ‘microrna 210’/dd OR ‘microrna 222’/dd OR ‘rna’/dd OR ‘small interfering rna’/dd OR ‘transcription factor’/dd) AND (‘congenital heart disease’/dm OR ‘diabetes mellitus’/dm OR ‘diabetic complication’/dm OR ‘intrauterine growth retardation’/dm OR ‘macrosomia’/dm OR ‘maternal diabetes mellitus’/dm OR ‘pregnancy diabetes mellitus’/dm)

**Table 2 jpm-13-01126-t002:** Data extracted from studies included in the systematic review.

Authors And Year of Publication	Study Design	Country or Continent	Sample Size	Mean Age	Gestational Time	Sample Type	miRNA Identification Technique	miRNA	Regulation	*p*-Value
[20]	Case-control	China	Case: 70 Control: 70	Case: 30.33 ± 2.25 Control: 30.04 ± 2.25	Case: 39.12 ± 1.15 Control: 38.76 ± 1.06 (weeks)	Blood	qRT-PCR	miR-134-5p	Up-regulation	<0.001
[21]	Case-control	China	Case: 26 Control: 23	Case: 30.6 ± 4.4 Control: 29.2 ± 3.5	Case: 38.1 ± 1.2 Control: 39.4 ± 1.2 (weeks)	Placenta	RT-PCR	miR-6869-5p	Down-regulation	<0.001
[22]	Case-control	China	Case: 110 Control: 78	Case: 33,118 ± 3402 Control: 32,385 ± 3579	Case: 25.591 ± 1.757 Control: 25.218 ± 2.196	Blood	qRT-PCR	miR-1323	Up-regulation	<0.05
[23]	Cross-sectional	China	Case: 36 Control: 37	20–40 Years	37–41 Weeks	Placenta	qRT-PCR	miR-125b; miR-144; miR-543	Up-regulation	<0.001
[24]	Case-control	China	Case: 30 Control: 30	Case: 24–39 Control: 24–39	24–28 Weeks	Plasma and Placenta	qRT-PCR	miR-345-3p	Down-regulation	<0.01
[25]	Case-control	Gana	Case: 137 Control: 158	Case: 30.94 ± 3.45 Control: 29.67 ± 3.64	Case: 272.22 ± 6.40 Control: 274.41 ± 8.02 (Days)	Placenta	qRT-PCR	miR-21	Down-regulation	<0.01
[26]	Case-control	China	Case: 68 Control: 55	Case:32.65 ± 4.63 Control: 31.27 ± 4.01	Case: 39.09 ± 1.11 Control: 38.98 ± 1.05 (weeks)	Plasma	qRT-PCR	miR-29a; miR-29b	Down-regulation	<0.001
[27]	Case-control	China	Case: 33 Control: 20	ND	Case: 24–26 weeks Control: 37–41 weeks	Placenta	qRT-PCR	miR-140-3p	Up-regulation	<0.05
[28]	Case-control	China	Case: 30 Control: 10	ND	ND	Blood	qRT-PCR	miR-330-3p	Up-regulation	<0.001
[29]	Case-control	China	Case: 20 Control: 20	Case: 29.4 ± 1.2 Control: 29.6 ± 0.4	Case: 38.7 ± 0.6 Control: 39.0 ± 1.0	Placenta	qRT-PCR	miR-17; miR-195; miR-257	Down-regulation; Up-regulation; Up-regulation	<0.01
[30]	Case-control	Turkey	Case: 19 Control: 28	Case: 30.4 ± 4.6 Control: 28.1 ± 5.8	Case: 33.5 ± 3.6 Control: 33 ± 4.1	Blood	qRT-PCR	miR-21-3p	Down-regulation	0.008
[31]	Case-control	Germany	Case: 30 Control: 30	Case: 31 ± 4 Control: 32 ± 4	Case: 27.0 ± 2.3 Control:27.6 ± 2.37	Blood	qRT-PCR	miRNA-340	Up-regulation	0.02
[32]	Cohort	United States	Case: 36 Control: 80	Case: 34.3 ± 3.6 Control: 32.9 ± 4.4	Case: 15.1 ± 2.9 Control: 16.5 ± 2.3	Blood	qRT-PCR	miR-21-3p; miR-155-5p	ND	0.005; 0.028
[33]	Case-control	Turkey	Case: 14 Control: 27	Case: 30.4 ± 4.4 Control: 27.9 ± 5.5	Case: 33.5 ± 3.5 Control: 33.1 ± 4.1	Blood	qPCR	miR-155-5p	Down-regulation	0.04
[34]	Case-control	China	Case: 30 Control: 38	Case: 30.27 ± 6.38 Control: 33.21 ± 8.17	Case: 38.12 ± 1.65 Control: 37.54 ± 1.31	Blood	qRT-PCR	miR-377-3p	Up-regulation	<0.01
[35]	Case-control	China	Case: 25 Control: 30	ND	ND	Blood	qRT-PCR	miR-181d	Up-regulation	<0.01
[36]	Case-control	Turkey	Case: 30 Control: 30	Case: 32.3 Control: 29.9	Case: 26.9 Control: 27.4	Plasma	qRT-PCR	miR-7-5p	Up-regulation	0.02
[37]	Case-control	China	Case: 20 Control: 20	Case: 35.30 ± 4.37 Control: 32.30 ± 3.40	Case: 74.38 ± 8.58 Control: 71.50 ± 11.72	Blood	qRT-PCR	miR-193b	Down-regulation	<0.001
[38]	Case-control	China	Case: 76 Control: 73	27.02 ± 3.17	ND	Blood	qRT-PCR	miR-409-5p	Up-regulation	<0.05
[39]	Cohort	Spain	Case: 31 Control: 29	Case: 31.9 ± 1.8 Control: 31.0 ± 3.6	Case: 39.1 ± 1.3 Control: 39.2 ± 1.2	Blood	qRT-PCR	miR-330-3p	Up-regulation	0.003
[40]	Case-control	China	Case: 10 Control: 10	Case: 30.03 ± 3.56 Control: 26.67 ± 4.59	Case: 17.66 ± 0.85 Control: 18.17 ± 0.93	Blood	qRT-PCR	miR-16-5p; miR-17-5p; miR-19a-3p; miR-19b-3p; miR-20a-5p	Up-regulation	5.36 × 10^−11^; 1.10 × 10^−10^; 6.57 × 10^−43^; 1.73 × 10^−74^; 5.27 × 10^−37^
[41]	Case-control	China	Case: 24 Control: 24	Case: 28.79 ± 2.21 Control: 29.46 ± 1.89	Case: 17.40 ± 0.70 Control: 17.16 ± 0.79	Plasma	qRT-PCR/ chip TLDA	miR-132; miR-29a; miR-222	Down-regulation	0.042; 0.032; 0.041
[42]	Case-control	China	Case: 100 Control: 100	ND	ND	Blood	qRT-PCR	miRNA-19a; miRNA-19b	Up-regulation	ND
[43]	Cohort	Canada	Case: 23 Control: 46	Case: 29.8 ± 5.3 Control: 27.9 ± 4.4	Case: 10.5 ± 2.5 Control: 10.6 ± 2.4	Blood	qRT-PCR	miR-122-5p; miR-132-3p; miR-1323; miR-136-5p; miR-182-3p; miR-210-3p; miR-29a-3p; miR-29b-3p; miR-342-3p; miR-520h	Up-regulation	0.01; 0.03; 0.03; 0.03; 0.01; 0.02; 0.03; 0.04; 0.008; 0.03
[44]	Case-control	China	Case: 48 Control: 46	Case: 29.86 ± 0.94 Control: 30.02 ± 0.89	ND	Placenta	qRT-PCR	miR-657	Up-regulation	<0.001
[45]	Case-control	China	Case: 30 Control: 29	Case: 28–40 Control: 29–38	Case: 37.9 ± 1.1 Control: 39.2 ± 1.1	Placenta	qRT-PCR	miR-657	Up-regulation	<0.01
[46]	Case-control	China	Case: 10 Control: 10	21–37 Years	ND	Placenta	qRT-PCR	miR-508-3p; miR-27a; miR-9; miR-137; miR-92a; miR-33a; miR-30d; miR-362-5p; miR-502-5p	Up-regulation; down-regulation; down-regulation; down-regulation; down-regulation; down-regulation; down-regulation; down-regulation; down-regulation	<0.01; <0.05; <0.05; <0.05; <0.05; <0.05; <0.05; <0.05; <0.05
[47]	Case-control	Europe	Case: 41 Control: 41	Case: 32.7 ± 4 Control: 33.2 ± 3.8	Case: 40 ± 39.3 Control: 40.1 ± 1.1	Blood	qRT-PCR	miR-16-5p; miR-29a-3p; miR-134-5p	Up-regulation	0.008; 0.004; 0.046
[48]	Case-control	China	Case: 25 Control: 25	ND	ND	Blood	qRT-PCR	miR-503	Up-regulation	<0.01
[49]	Case-control	China	Case: 20 Control: 20	ND	ND	Blood	qRT-PCR	miR-494	Down-regulation	<0.01
[50]	Cohort	Italy	Case: 21 Control: 10	Case: 35.57 ± 5.63 Control: 32.80 ± 5.16	ND	Blood	qRT-PCR	miR-330-3p	Up-regulation	0.01
[51]	Cohort	Italy	Case: 12 Control: 12	Case: 36.4 ± 4.6 Control: 34.9 ± 5.1	ND	Plasma	qRT-PCR	miR-222-3p; miR-409-3p	Up-regulation	0.05; 0.01
[52]	Case-control	China	Case: 123 Control: 123	Case: 31.23 ± 2.31 Control: 30.84 ± 2.95	Case: 39.35 ± 3.81 Control: 39.52 ± 2.47	Plasma	qRT-PCR	miR-96-5p	Down-regulation	0.01
[53]	Case-control	China	Case: 108 Control: 50	Case: 30.89 ± 3.45 Control: 30.10 ± 2.79	Case: 25.20 ± 1.23 Control: 25.32 ± 1.45	Blood and placenta	qRT-PCR	miR-132	Down-regulation	<0.001
[54]	Case-control	Egypt	Case: 109 Control: 103	Case: 29.9 ± 6.28 Control: 29.7 ± 5.85	ND	Blood	qRT-PCR	miR-223	Up-regulation	<0.001
[55]	Case-control	China	Case: 13 Control: 13	Case: 27.62 ± 3.10 Control: 27.85 ± 3.36	ND	Adipose tissue	qRT-PCR	miR-222	Up-regulation	<0.01
[56]	Case-control	China	Case: 30 Control: 26	Case: 27.6 Control: 26.9	ND	Blood	qRT-PCR	miR-214	Down-regulation	<0.05
[57]	Case-control	China	Case: 204 Control: 202	Case: 30.91 ± 0.38 Control: 29.68 ± 0.53	Case: 271.68 ± 11.47 Control: 274.37 ± 8.10 (days)	Placenta	qRT-PCR	miR-29b	Down-regulation	<0.05
[58]	Case-control	Spain and Italy	Case: 23 Control: 20	Case: 34.0 (32.5–37.5) Control: 34.5 (32.0–37.2)	Case: 40.3 (38.4–40.7) Control: 39.9 (39.4–40.7)	Plasma	qRT-PCR	miR-223; miR-23a	Up-regulation	1.4 × 10^−7^; 0.019
[59]	Case-control	China	Case: 30 Control: 30	ND	ND	Blood	qRT-PCR	miR-770-5p	Up-regulation	<0.01
[60]	Case-control	China	Case: 32 Control: 48	Case: 32.71 ± 5.26 Control: 29.13 ± 4.22	Case: 28.33 ± 2.81 Control: 29.10 ± 2.32	Blood	qRT-PCR	miR-520h	Up-regulation	<0.001
[61]	Case-control	China	Case: 5 Control: 5	ND	ND	Placenta	qRT-PCR	miR-9-5p	Down-regulation	<0.01
[62]	Case-control	South Africa	Case: 28 Control: 53	Case: 29.5 ± 6.2 Control: 28.6 ± 6.4	ND	Blood	qRT-PCR	miR-20a-5p; miR-222-3p	Down-regulation	0.038; 0.027
[63]	Case-control	China	Case: 12 Control: 12	ND	ND	Blood	qRT-PCR	miR-33a-5p	Up-regulation	<0.01
[64]	Case-control	China	Case: 112 Control: 58	Case: 31.59 ± 3.93 Control: 31.14 ± 3.94	Case: 24.85 ± 1.69 Control: 24.47 ± 2.15	Blood and placenta	qRT-PCR	miR-136	Up-regulation	<0.001
[65]	Case-control	China	Case: 166 Control: 196	Case: 31.03 ± 3.63 Control: 29.70 ± 3.83	ND	Placenta	qRT-PCR	miR-30d-5p	Down-regulation	<0.01
[66]	Case-control	China	Case: 102 Control:102	Case: 29.8 ± 3.2 Control: 29.5 ± 2.8	Case: 27.0 ± 1.6 Control: 26.8 ± 1.1	Blood	qRT-PCR	miR-195-5p	Up-regulation	<0.01
[67]	Case-control	China	Case: 193 Control: 202	Case: 30.93 ± 3.45 Control: 29.68 ± 3.66	Case: 272.22 ± 6.39 Control: 274.41 ± 8.03 (days)	Placenta	qRT-PCR	miR-98	Up-regulation	<0.05
[68]	Case-control	China	Case: 85 Control: 72	Case: 26.8 ± 3.5 Control: 26.4 ± 3.6	Case: 25.8 ± 2.5 Control: 26.1 ± 1.2	Blood	qRT-PCR	miR-16-5p; miR-17-5p; miR-20a-5p	Up-regulation	<0.01
[69]	Case-control	China	Case: 53 Control: 46	Case: 29.8 ± 0.4 Control: 29.2 ± 0.6	Case: 39.9 ± 0.1 Control: 39.0 ± 0.2	Blood	qRT-PCR	miR-574-5p; miR-3135b	Down-regulation	<0.0001; 0.002
[70]	Case-control	Mexico	Case: 27 Control: 34	Case: 29.93 ± 6.03 Control: 26.06 ± 5.28	ND	Urine (urinary exosome)	qRT-PCR	miR-516-5p; miR-517-3p; miR-518-5p; miR-222-3p; miR-16-5p	Down-regulation; down-regulation; down-regulation; down-regulation; up-regulation	<0.05; <0.05; <0.01; <0.01; <0.01
[71]	Case-control	China	Case: 11 Control: 12	Case: 29.91 ± 1.385 Control: 29.08 ± 1.305	ND	Blood	qRT-PCR	miR-137	Up-regulation	0.0073
[72]	Case-control	Estonia	Case: 13 Control: 9	Case: 31.1 ± 4.2 Control: 28.1 ± 4.5	23–31 weeks	Blood	qRT-PCR	let-7e-5p; let-7g-5p; miR-100-5p; miR-101-3p; miR-146a-5p; miR-195-5p; miR-222-3p; miR-23b-3p; miR-30b-5p; miR-30c-5p; miR-30d-5p; miR-342-3p; miR-423-5p	Up-regulation	0.03; 0.01; 0.04; 0.03; 0.03; 0.03; 0.03; 0.02; 0.04; 0.02; 0.03; 0.04; 0.02
[73]	Case-control	Iran	Case: 30 Control: 30	ND	ND	Plasma	qRT-PCR	miR-135a	Down-regulation	0.001
[74]	Case-control	China	Case: 5 Control: 5	ND	ND	Placenta	qRT-PCR	miR-190b	Up-regulation	<0.001

ND: Not described; qRT-PCR: Polymerase chain reaction via quantitative reverse transcriptase.

**Table 3 jpm-13-01126-t003:** Summary of responses for each study included in the risk of bias assessment.

Reference	Q1	Q2	Q3	Q4	Q5	Q6	Q7	Q8	Q9	Q10	Q11
[20] ^1^	Y	Y	Y	Y	Y	N	NA	Y	Y	Y	NA
[21] ^1^	Y	Y	Y	Y	Y	N	NA	Y	Y	Y	NA
[22] ^1^	Y	Y	Y	Y	Y	N	NA	Y	Y	Y	NA
[23] ^3^	Y	Y	Y	Y	N	NA	Y	Y	NA	NA	NA
[24] ^1^	Y	Y	Y	Y	Y	N	NA	Y	Y	Y	NA
[25] ^1^	Y	Y	Y	Y	Y	N	NA	Y	Y	Y	NA
[26] ^1^	Y	Y	Y	Y	Y	N	NA	Y	Y	Y	NA
[27] ^1^	Y	Y	Y	Y	Y	Y	Y	Y	Y	Y	NA
[28] ^1^	Y	Y	Y	Y	Y	N	NA	Y	Y	Y	NA
[29] ^1^	Y	Y	Y	Y	Y	N	NA	Y	Y	Y	NA
[30] ^1^	Y	Y	Y	Y	Y	N	NA	Y	Y	Y	NA
[31] ^1^	Y	Y	Y	Y	Y	Y	Y	Y	Y	Y	NA
[32] ^2^	Y	Y	Y	N	NA	Y	Y	Y	Y	Y	Y
[33] ^1^	Y	Y	Y	Y	Y	Y	Y	Y	Y	Y	NA
[34] ^1^	Y	Y	Y	Y	Y	N	NA	Y	Y	Y	NA
[35] ^1^	Y	Y	Y	Y	Y	N	NA	Y	Y	Y	NA
[36] ^1^	Y	Y	Y	Y	Y	N	NA	Y	Y	Y	NA
[37] ^1^	Y	Y	Y	Y	Y	N	NA	Y	Y	Y	NA
[38] ^1^	Y	Y	Y	Y	Y	N	NA	Y	Y	Y	NA
[39] ^2^	Y	Y	Y	N	NA	Y	Y	Y	Y	Y	Y
[40] ^1^	Y	Y	Y	Y	Y	Y	Y	Y	Y	Y	NA
[41] ^1^	Y	Y	Y	Y	Y	Y	Y	Y	Y	Y	NA
[42] ^1^	Y	Y	Y	Y	Y	N	NA	Y	Y	Y	NA
[43] ^2^	Y	Y	Y	Y	Y	N	Y	Y	Y	Y	Y
[44] ^1^	Y	Y	Y	Y	Y	Y	Y	Y	Y	Y	NA
[45] ^1^	Y	Y	Y	Y	Y	N	NA	Y	Y	Y	NA
[46] ^1^	Y	Y	Y	Y	Y	N	NA	Y	Y	Y	NA
[47] ^2^	Y	Y	Y	Y	Y	N	Y	Y	Y	Y	Y
[48] ^1^	Y	Y	Y	Y	Y	N	NA	Y	Y	Y	NA
[49] ^1^	Y	Y	Y	Y	Y	N	NA	Y	Y	Y	NA
[50] ^2^	Y	Y	Y	Y	Y	N	Y	Y	Y	Y	Y
[51] ^2^	Y	Y	Y	N	NA	Y	Y	Y	Y	Y	Y
[52] ^1^	Y	Y	Y	Y	Y	N	NA	Y	Y	Y	NA
[53] ^1^	Y	Y	Y	Y	Y	N	NA	Y	Y	Y	NA
[54] ^1^	Y	Y	Y	Y	Y	N	NA	Y	Y	Y	NA
[55] ^1^	Y	Y	Y	Y	Y	N	NA	Y	Y	Y	NA
[56] ^1^	Y	Y	Y	Y	Y	N	NA	Y	Y	Y	NA
[57] ^1^	Y	Y	Y	Y	Y	N	NA	Y	Y	Y	NA
[58] ^1^	Y	Y	Y	Y	Y	N	NA	Y	Y	Y	NA
[59] ^1^	Y	Y	Y	Y	Y	N	NA	Y	Y	Y	NA
[60] ^1^	Y	Y	Y	Y	Y	N	NA	Y	Y	Y	NA
[61] ^1^	Y	N	Y	Y	Y	N	NA	Y	Y	Y	NA
[62] ^1^	Y	Y	Y	Y	Y	N	NA	Y	Y	Y	NA
[63] ^1^	Y	Y	Y	Y	Y	N	NA	Y	Y	Y	NA
[64] ^1^	Y	Y	Y	Y	Y	N	NA	Y	Y	Y	NA
[65] ^1^	Y	Y	Y	Y	Y	N	NA	Y	Y	Y	NA
[66] ^1^	Y	Y	Y	Y	Y	Y	Y	Y	Y	Y	NA
[67] ^1^	Y	Y	Y	Y	Y	N	NA	Y	Y	Y	NA
[68] ^1^	Y	Y	Y	Y	Y	N	NA	Y	Y	Y	NA
[69] ^1^	Y	Y	Y	Y	Y	N	NA	Y	Y	Y	NA
[70] ^1^	Y	Y	Y	Y	Y	N	NA	Y	Y	Y	NA
[71] ^1^	Y	Y	Y	Y	Y	N	NA	Y	Y	Y	NA
[72] ^1^	Y	Y	Y	Y	Y	N	NA	Y	Y	Y	NA
[73] ^1^	Y	N	Y	Y	Y	N	NA	Y	Y	Y	NA
[74] ^1^	Y	Y	Y	Y	Y	N	NA	Y	Y	Y	NA

^1^ Case-control; ^2^ Cohort; ^3^ Cross-sectional. Y—Yes; N—No; NA—Not applicable; U—Unclear.

**Table 4 jpm-13-01126-t004:** Biological function and regulation of miRNAs found in the systematic review.

miRNA	Role/Biological Function	Regulation	Reference
let-7e-5p	Osteoblast differentiation and bone formation	↑	[72]
let-7g-5p	Inhibit mammary cell proliferation	↑	[72]
miR-7-5p	Regulates insulin sensitivity	↑	[36]
miR-9	Mammalian neuronal development and function	↓	[46]
miR-9-5p	Regulation of aerobic glycolysis and mitochondrial complex expression	↓	[61]
miR-16-5p	It targets insulin receptor substrate (IRS) proteins 1 and 2 and mediates insulin-like growth factor-I (IGF-I), which is closely related to insulin resistance	↑	[40,47,68,70]
miR-17	Anti-diabetic activity by the anti-inflammation effect on macrophage	↓	[29]
miR-17-5p	Improves insulin and blood glucose metabolic imbalances	↑	[40,68]
miR-19a	Promotes cell proliferation and angiogenesis by regulating the PI3K/AKT pathway, known as insulin signaling pathway	↑	[42]
miR-19a-3p	Regulates insulin secretion while it suppresses the apoptosis of pancreatic β cells	↑	[40]
miR-19b	Regulates cells apoptosis	↑	[42]
miR-19b-3p	Regulates the immune and nervous system	↑	[40]
miR-20a-5p	Improves glucose uptake and relieves diabetic cardiac hypertrophy, fibrosis, inflammation, and apoptosis	↑	[40,68]
		↓	[62]
miR-21	Keeps glucose levels down in patients with diabetic complications	↓	[25]
miR-21-3p	Regulates the healing process of diabetic lesions	↓	[30]
		ND	[32]
miR-23a	Regulates the inflammatory response	↑	[58]
miR-23b-3p	Promotes high glucose-induced cellular metabolic memory in diabetic retinopathy	↑	[72]
miR-27a	Promotes insulin resistance	↓	[46]
miR-29a	Acts as an important regulator of insulin-stimulated glucose metabolism	↓	[26,41]
miR-29a-3p	Regulator of insulin-like growth factor 1 receptor	↑	[43,47]
miR-29b	Regulates aortic remodeling and stiffening in diabetes	↓	[26,57]
miR-29b-3p	Promotes cell proliferation, apoptosis, fibrosis, and inflammation	↑	[43]
miR-30b-5p	Induces abnormal angiogenesis	↑	[72]
miR-30c-5p	Induces macrophage-mediated inflammation and pro-atherosclerosis signal pathways	↑	[72]
miR-30d	Induces insulin production	↓	[46]
miR-30d-5p	Regulates human chorionic trophoblast cell proliferation, migration, and glucose uptake capacity	↑	[72]
		↓	[65]
miR-33a	Regulates the insulin signaling pathway and glucose metabolism	↓	[46]
miR-33a-5p	Related to decreased cell growth (β cell) and insulin production	↑	[63]
miR-92a	Modulation of vascular homeostasis	↓	[46]
miR-96-5p	Relieves the antiproliferative effects induced by high glucose conditions on trophoblasts	↓	[52]
miR-98	Protects nucleus pulposus cells against apoptosis	↑	[67]
miR-100-5p	Promotes angiogenesis during the implantation process	↑	[72]
miR-101-3p	Associated with insulin production, survival, and death of pancreatic β cells	↑	[72]
miR-122-5p	Suppresses insulin resistance	↑	[43]
miR-125b	Modulates insulin secretion	↑	[23]
miR-132	Promote the trophoblast cell proliferation	↓	[41,53]
miR-132-3p	Related to insulin resistance and β cell function	↑	[43]
miR-134-5p	Regulate trophoblast cell behaviors in preeclampsia, such as apoptosis, migration, and invasion	↑	[20,47]
miR-135a	Related to the proliferation and migration of cells	↓	[73]
miR-136	Relieves inhibited cell viability induced by high glucose treatment	↑	[64]
miR-136-5p	Related to cellular proliferation, migration, and invasion	↑	[43]
miR-137	Participates in the interaction between endothelial cells and monocytes	↑	[71]
		↓	[46]
miR-140-3p	Important regulator of osteoblastogenesis	↑	[27]
miR-144	Regulates proliferation and apoptosis	↑	[23]
miR-146a-5p	Inhibit the inflammatory response and apoptosis	↑	[72]
miR-155-5p	Promotes fibrosis of proximal tubule cells	↓	[33]
		ND	[32]
miR-181d	Modulates the process of insulin signaling, cell viability, and apoptosis in pancreatic β cells	↑	[35]
miR-182-3p	Related to vascular smooth muscle cells proliferation and migration	↑	[43]
miR-190b	Inhibits pancreatic β cell proliferation and insulin secretion	↑	[74]
miR-193b	Suppresses apoptosis and autophagy	↓	[37]
miR-195	Increases epithelial-mesenchymal transition and cell permeability	↑	[29]
miR-195-5p	Relates to cell viability and proliferation, and apoptosis	↑	[66,72]
miR-210-3p	Reduces cell viability and promotes apoptosis of pancreatic β cells	↑	[43]
miR-214	Involved in cellular immune response, regulating cell activation, proliferation, and differentiation	↓	[56]
miR-222	Regulates estrogen receptor (ER) and glucose transporter 4 (GLUT4)	↑	[55]
		↓	[41]
miR-222-3p	Relates to glucose metabolism in pregnancy	↑	[51,72]
		↓	[62,70]
miR-223	Modulator of human myeloid differentiation	↑	[54,58]
miR-257	ND	↑	[29]
miR-330-3p	Involved in insulin secretion and β cell growth and proliferation, activating proteins involved in cell cycle and cell growth	↑	[28,39,50]
miR-340	Associated with induced hyperinsulinemia	↑	[31]
miR-342-3p	Induction of vascular dysfunction	↑	[43,72]
miR-345-3p	Regulates cell growth, apoptosis, migration, and invasion	↓	[24]
miR-362-5p	Promotes cell proliferation and inhibition of apoptosis	↓	[46]
miR-377-3p	Regulates VEGF expression	↑	[34]
miR-409-3p	Relates to immune dysregulation in autoimmune diabetes	↑	[51]
miR-409-5p	Correlated with insulin resistance	↑	[38]
miR-423-5p	Relieves high glucose-mediated podocyte injuries	↑	[72]
miR-494	Decreased insulin secretion and cell proliferation, and increased apoptosis	↓	[49]
miR-502-5p	Mediates insulin resistance	↓	[46]
miR-503	Impairs restorative angiogenesis	↑	[48]
miR-508-3p	Promotes cell proliferation and inhibits apoptosis	↑	[46]
miR-516-5p	Inhibits cell proliferation	↓	[70]
miR-517-3p	Promotes trophoblast dysfunction	↓	[70]
miR-518-5p	Increases hypoxia-induced vascular endothelial cell damage	↓	[70]
miR-520h	Inhibit cell viability and promote cell apoptosis	↑	[43,60]
miR-543	Regulates high glucose-induced fibrosis and autophagy	↑	[23]
miR-574-5p	Metabolic regulator for serum lipids and blood glucose	↓	[69]
miR-657	Promotes macrophage polarization	↑	[44,45]
miR-770-5p	Inhibits cell proliferation and promotes apoptosis	↑	[59]
miR-1323	Promotes the trophoblast cell viability under high glucose conditions	↑	[22,43]
miR-3135b	Metabolic regulator for serum lipids and blood glucose	↓	[69]
miR-6869-5p	Involved in maintaining the microenvironmental balance of the placenta	↓	[21]

ND: not described; ↑: up-regulated; ↓: down-regulated.

## Data Availability

No new data were created or analyzed in this study. Data sharing does not apply to this article.

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
