# Peer review of "MicroRNAs Associated with the Pathophysiological Mechanisms of Gestational Diabetes Mellitus: A Systematic Review for Building a Panel of miRNAs"

_jpm, 2023, doi:10.3390/jpm13071126_

Round 1
Reviewer 1 Report
It is considered an interesting systematic review of GDM and miRNA.
However, it would be nice to explain the research method in a little detail.
1. From when to when was the publication period of the literatures searched?
2. It would be nice if authirs could explain whether only the literature in the research field of the literature, that is, in the medical field, was included in the search, or whether the literature in other academic fields was also included in the search.
It would be nice to correct some sentences to suit academic papers.
Author Response
Dear Reviewer,
We would like to thank you, the reviewers, and the editorial board for their important comments regarding our manuscript entitled “MicroRNAs associated with the pathophysiological mechanisms of Gestational Diabetes Mellitus: A systematic review for building a panel of miRNAs”, ID JPM-2479034, and for the opportunity to improve our manuscript. We have fully revised our manuscript, incorporated the suggestions, and answered each comment point-by-point. We do believe that we have consistently answered the reviewers’ comments and that the new version of the manuscript has achieved higher quality to be considered for publication.
We look forward to hearing from you.
Sincerely,
A.AS.R.
ID JPM-2479034 - MicroRNAs associated with the pathophysiological mechanisms of Gestational Diabetes Mellitus: A systematic review for building a panel of miRNAs
Answers to the editor requirements:
REVIEWER 1:
|
1 |
From when to when was the publication period of the literatures searched? |
|
The selected studies were published between 2011 and 2022. This information was added to the paper. |
|
|
2 |
It would be nice if authors could explain whether only the literature in the research field of the literature, that is, in the medical field, was included in the search, or whether the literature in other academic fields was also included in the search. |
|
Done. Only observational studies with original research in the area of human and medical genetics were included in our systematic review. This information was added to the manuscript. |
|

Reviewer 2 Report
The authors have provided an excellent summary of different micro RNAs involved in Gestational diabetes mellitus (GDM). The review is well-written and well-organized. I think considering the following aspects in the review will provide more strength to the content.
Circulating micro RNAs presented in the study possibly originated from blood cells and different bodily tissues, including the placenta. As mentioned in the introduction, the micro RNAs from the placenta could be key in regulating hormones and further leading to GDM; In this regard, It is required to emphasize and discuss the specific role of microRNAs identified in body fluids that has placental origin, which are more relevant biomarkers to detect GDM. One example, miR-520h originated in the placenta because it is present only in blood samples of pregnant subjects and completely absent in nonpregnant controls (Kotlabova et al., 2011). The current review demonstrates that miR-520h is increasingly present in blood samples of GDM subjects.
Kotlabova, Katerina, Jindrich Doucha, and Ilona Hromadnikova. "Placental-specific microRNA in maternal circulation–identification of appropriate pregnancy-associated microRNAs with diagnostic potential." Journal of reproductive immunology 89.2 (2011): 185-191.
In the results section, the authors presented evidence of the detection of circulating micro RNAs in body fluids from different studies. In the discussion part, to support the role of the same micro RNAs contribution to GDM, the authors presented evidence of these micro RNAs regulating the post-gene transcription in different tissues and cells. For example, the authors mentioned that mir-330-3p is increasingly present in blood samples of GDM subjects, according to references 28, 39, and 50. On the other hand, in the discussion part, the impact of mir-330-3p in regulating the expression of different genes is presented in pancreatic beta cells. Keeping in mind of the general audience of all disciplines, It is worth mentioning or discussing the possibility and the pathways by which miRNAs are translocated between their origin, source, and targeting tissue.
Lane 202-206, it was stated that the mir-16-5p is the modulator of the PI3K/Akt signaling pathway, and overexpression of these pathway genes (Pi3Kr1, Pi3kr3, mTOR, and Mapk3) is associated with the development of diabetes. However, based on the reference provided (Kwon et al. 2014), Cmah knock-out mice that develop diabetes has increased levels of mir-16 and low levels of Pi3K and Mapk3. It is expected that Increased mir-16 degrades pi3k and mapk3 mRNAs in knout mice, which triggers T2D. Hence, the overexpression of these genes might not be related to the development of diabetes, as stated. The statement requires appropriate corrections.
Author Response
Dear Reviewer,
We would like to thank you, the reviewers, and the editorial board for their important comments regarding our manuscript entitled “MicroRNAs associated with the pathophysiological mechanisms of Gestational Diabetes Mellitus: A systematic review for building a panel of miRNAs”, ID JPM-2479034, and for the opportunity to improve our manuscript. We have fully revised our manuscript, incorporated the suggestions, and answered each comment point-by-point. We do believe that we have consistently answered the reviewers’ comments and that the new version of the manuscript has achieved higher quality to be considered for publication.
We look forward to hearing from you.
Sincerely,
A.A.S.R.
ID JPM-2479034 - MicroRNAs associated with the pathophysiological mechanisms of Gestational Diabetes Mellitus: A systematic review for building a panel of miRNAs
Answers to the editor requirements:
REVIEWER 2:
|
1 |
Circulating micro RNAs presented in the study possibly originated from blood cells and different bodily tissues, including the placenta. As mentioned in the introduction, the micro RNAs from the placenta could be key in regulating hormones and further leading to GDM; In this regard, It is required to emphasize and discuss the specific role of microRNAs identified in body fluids that has placental origin, which are more relevant biomarkers to detect GDM. One example, miR-520h originated in the placenta because it is present only in blood samples of pregnant subjects and completely absent in nonpregnant controls (Kotlabova et al., 2011). The current review demonstrates that miR-520h is increasingly present in blood samples of GDM subjects. Kotlabova, Katerina, Jindrich Doucha, and Ilona Hromadnikova. "Placental-specific microRNA in maternal circulation–identification of appropriate pregnancy-associated microRNAs with diagnostic potential." Journal of reproductive immunology 89.2 (2011): 185-191. |
|
Done. We added the following paragraph to discuss the role of miRNAs with placental origin:
The placenta produces several miRNAs expressed specifically by placental cells. They can be dysregulated in the plasm and placenta of women with GDM, being also associated with pregnancy and birth-related outcomes. Detection of placental miRNAs in placental cells and circulation contribute to the understanding of the molecular pathways and intracellular signalization and their influences on GDM genetic background [86]. |
|
|
2 |
In the results section, the authors presented evidence of the detection of circulating micro RNAs in body fluids from different studies. In the discussion part, to support the role of the same micro RNAs contribution to GDM, the authors presented evidence of these micro RNAs regulating the post-gene transcription in different tissues and cells. For example, the authors mentioned that mir-330-3p is increasingly present in blood samples of GDM subjects, according to references 28, 39, and 50. On the other hand, in the discussion part, the impact of mir-330-3p in regulating the expression of different genes is presented in pancreatic beta cells. Keeping in mind of the general audience of all disciplines, It is worth mentioning or discussing the possibility and the pathways by which miRNAs are translocated between their origin, source, and targeting tissue. |
|
Done. The following paragraph was add to discuss the topic:
Due to the relation between miRNAs and the regulation of gene expression, several information about the tissue origin of malignant cells could be generated. Since their initial discovery, it has been clear that miRNAs are expressed in a variety of cell types and that their expression patterns are tissue-specific, and thus could have great diagnostic importance and prognostic value [85]. |
|
|
3 |
Lane 202-206, it was stated that the mir-16-5p is the modulator of the PI3K/Akt signaling pathway, and overexpression of these pathway genes (Pi3Kr1, Pi3kr3, mTOR, and Mapk3) is associated with the development of diabetes. However, based on the reference provided (Kwon et al. 2014), Cmah knock-out mice that develop diabetes has increased levels of mir-16 and low levels of Pi3K and Mapk3. It is expected that Increased mir-16 degrades pi3k and mapk3 mRNAs in knout mice, which triggers T2D. Hence, the overexpression of these genes might not be related to the development of diabetes, as stated. The statement requires appropriate corrections. |
|
Done. The statement has been corrected and the following text was added:
Kwon et al. [76] observed that the upregulation of mir-16-5p in the liver of Cmah-null mice, used as models for T2DM, can negatively regulate the insulin/PI3K-AKT signaling pathway in association with other genes. These results suggest that the impairment of insulin mechanisms may favor the development of metabolic disorders, such as chronic hyperglycemia. |
|